# Epidemiological Features and Risk Factors for Acquiring Hepatitis B, Hepatitis C, and Syphilis in HIV-Infected Patients in Shaanxi Province, Northwest China

**DOI:** 10.3390/ijerph17061990

**Published:** 2020-03-18

**Authors:** Chao Zhang, Qiang Ren, Wenhui Chang

**Affiliations:** Department of AIDS Prevention and Control, Shaanxi Provincial Center for Disease Control and Prevention, Xi’an 710054, China; zhangchao3759@126.com (C.Z.); renqiang2514@126.com (Q.R.)

**Keywords:** HIV-positive individuals, Hepatitis B virus (HBV), Hepatitis C virus (HCV), syphilis, co-infection

## Abstract

Human immunodeficiency virus (HIV)-infected patients are at a higher risk for co-infection with Hepatitis B virus (HBV), Hepatitis C virus (HCV), and Treponema pallidum (TP; the agent causing syphilis) than the general population. The prevalence of HBV, HCV, and syphilis has geographic differences and varies from region to region among HIV-positive individuals. A retrospective study was carried out on HIV-positive individuals between June 2011 and June 2016 in Shaanxi Province. Univariate and multivariate logistic regression analyses using stepwise regression analysis regarding risk factors for HIV–HBV, HIV–HCV, and HIV–syphilis co-infection. HBV–HCV, HCV–syphilis, HBV–syphilis, and HBV–HCV–syphilis co-infection rates were 1.7%, 2.2%, 2.6%, and 0.1%, respectively. The rate of ineffective hepatitis B vaccine immunization was as high as 30.2% among HIV-positive individuals. Ethnicity (OR = 31.030, 95% CI: 11.643–82.694) and HIV transmission routes (OR = 134.024, 95% CI: 14.328–1253.653) were the risk factors for HCV infection in HIV-positive individuals. Among the HIV-positive individuals with the antibodies of TP, the rate of homosexual transmission was also higher, but heterosexual transmission was lower (OR = 0.549 95% CI: 0.382–0.789) The HIV-infected patients in Shaanxi Province had the characteristics of low active detection rate and late diagnosis. The high rate of ineffective vaccination against HBV suggests a need for improved vaccination services.

## 1. Introduction

Human immunodeficiency virus (HIV) shares similar routes of transmission with Hepatitis B virus (HBV), Hepatitis C virus (HCV), and Treponema pallidum (TP; the agent causing syphilis), which include blood transfusion, sexual transmission, and mother-to-child transmission. Therefore, it is very common that HIV-positive individuals are co-infected with HBV, HCV, and syphilis. It is estimated that approximately 5–20% of HIV-positive individuals worldwide are infected with HBV [1], and 15–30% are simultaneously infected with HCV [2]. Similarly, syphilis has been widely spread among HIV-positive individuals in the last few years, especially in men who have sex with men (MSM) [3,4]. There is no doubt that the high rate of HBV, HCV, and syphilis among HIV-positive individuals has become a global public health problem [5,6].

Increasing morbidity and mortality have been observed in a growing number of co-infected cases, although a single infection with any of the pathogens can cause serious health problems [7]. Co-morbidities such as chronic liver disease, which are caused by HBV and HCV, are considered to be serious problems among HIV-positive individuals [8]. There is evidence that co-infection with HBV or HCV can adversely affect the clinical process of HIV infection [9,10]. HIV–HBV co-infected individuals, especially those who have high HBV DNA viral load, are associated with lower CD4+ T-cell counts before the treatment [11]. Similarly, several studies have found that poorer immunological response to antiretroviral therapy (ART) in HIV–HCV co-infected individuals, and co-infection with HCV also increases the morbidity and mortality among HIV-positive individuals [12,13]. In addition, the direct-acting antivirals (DAAs) to HCV show serious drug–drug interactions with antiretroviral agents, that makes the treatment of both diseases more complicated [14,15]. Conversely, HIV infections are able to accelerate the process of hepatitis B and hepatitis C, leading to faster development to fibrosis and cirrhosis [16,17], as well as make liver disease one of the most important non-AIDS (non-acquired immune deficiency syndrome) causes of death in HIV-infected individuals in the past few years [18]. In recent years, a resurgence of TP has occurred in China [19]. As an ulcerative infection, syphilis not only increases HIV viral load and decreases CD4+ T-cell counts in HIV-positive individuals [20], but also increases the risk of sexually transmitted infections (STIs) [21]. In a word, mixed infection not only causes changes in biological behavior between viruses, but also affects each other’s natural course, complicates the clinical manifestations of infected individuals, and ultimately brings difficulties to clinicians’ diagnosis and treatment [22,23,24]. It also brings individuals, families, and society heavy burden, especially in underdeveloped areas.

In China, HIV-positive individuals are usually found late, especially in the elderly [25]. Most are discovered by means of passive detection, such as preoperative testing, other patient detection, and sexually transmitted diseases (STDs) clinics, etc. [26]. Due to common routes of transmission, patients may not understand their infection status of HBV, HCV, and syphilis, which may accelerate the progression of HIV severely. Moreover, studies show that the rates of co-infection with HBV, HCV, and syphilis have geographical differences and vary from region to region [27,28,29]. In China, studies reported that, 5.3–19.4% were positive for HBV in HIV-positive individuals [30,31,32,33,34], monitored by Hepatitis B surface antigen (HBsAg), while HCV positivity (surveilled by anti-HCV antibodies) ranging between 2.2% and 62.4% [30,31,32,33,34,35]. However, the epidemiological features of HBV, HCV, and syphilis infection in HIV-positive populations remain unclear in many parts of China. In addition, the impact of HBV, HCV, and syphilis on immunological indicators of HIV-positive individuals is poorly characterized. Therefore, the purpose of this study was to investigate the epidemiological features and risk factors of HBV, HCV, and syphilis infection among HIV-positive individuals in Shaanxi Province, which is less developed for economics in northwest China, and their CD4+ T-cell levels at baseline.

## 2. Materials and Methods

### 2.1. Study Population

A retrospective study was conducted using blood samples collected from 1018 HIV-positive individuals whose positive status was confirmed by western blotting (HIV Blot 2.2 WB, MP Biomedicals Asia Pacific Pte. Ltd., Singapore) between June 2011 and June 2016 at Shaanxi Provincial Center for Disease Prevention and Control. All participants in this study were HIV-1 seropositive. The epidemiological data of the subjects were collected from the national HIV/AIDS information system in China. Individuals were defined as having a HBV infection if HBsAg was positive. Similarly, anti-HCV antibody positive was defined as current or resolved HCV infection, and anti-TP antibody positive was defined as syphilis infection.

### 2.2. Enzyme-linked Immunosorbent Assay (ELISA)

Serum samples were used for HbsAg (Hepatitis B surface antigen), anti-HbsAb (Hepatitis B surface antibody), HbeAg (Hepatitis B “e” antigen), anti-Hbe (Hepatitis B e antibody), anti-HBc (Hepatitis B core antibody), anti-HCV (HCV antibody) and anti-TP (Treponema Pallidum antibody) detection by ELISA (Lizhu Co., China). Limiting antigen avidity enzyme immunoassay (LAg-avidity EIA; Jinhao Co., China) was employed to detect recent HIV infection. All ELISA was carried out according to the manufacturer’s instructions.

### 2.3. CD4+ T-Cell Count

The CD4+ T-cell counts were measured by the flow cytometry (BD Trucount Tubes) using the FACSCalibur apparatus (Becton–Dickinson, Franklin Lakes, NJ, USA) and the results were expressed in cells/mm^3^.

### 2.4. Estimation of HIV Infection Time and Age

The epidemiological data and CD4+ T-cell counts within one year of the report were collected and analyzed. According to the rule that the CD4+ T-cell counts were reduced by 50/μL per year after infection with HIV, the possible infection time and age were presumed [25,36]. Specific rules were shown in Table 1.

### 2.5. Statistical Analyses

The rates of HBV, HCV, and syphilis co-infections among HIV-positive individuals were analyzed by descriptive statistics and are presented as percentages. The relationship between categorical variables was determined by Chi-squared and Fisher’s exact tests. Risk factors were determined by univariate and multivariate logistic regression analysis using stepwise regression analysis. Odds ratio (OR) with 95% confidence interval and *p*-value were calculated. Student’s *t*-test was used for the comparisons of continuous variables between groups. All data were analyzed by SPSS 25.0 Software (SPSS Inc., Chicago, Illinois, USA) and a *p*-value <0.05 was considered significant.

### 2.6. Ethical Approval

This study received ethical approval from the Ethics Committee of Shaanxi Provincial Center for Disease Control and Prevention (No. 036). All experiments performed in this study were in accordance with the national laws and regulations of China.

## 3. Results

### 3.1. Sociodemographic Characteristics of HIV-Infected Patients

The retrospective analysis of laboratory records identified 1018 HIV-positive individuals between June 2011 and June 2016 at Shaanxi Provincial Center for Disease Control and Prevention, whose average age was 36.0 ± 13.2 years old, ranging from 2 to 83 years old. And their median age was 34.0 years old. The age distribution was mainly concentrated in 20–50 years old, accounting for 80.6%. Ethnicity groups of the 1018 HIV-positive individuals included Han Nationality (91.5%), Yi Nationality (7.0%), and others (1.6%). A large proportion of individuals (46.8%) were married with spouse. More than 64.1% of patients had low literacy (junior high school level and below). HIV infection was mainly transmitted through sexual contact (90.7%), including heterosexual transmission (65.5%) and homosexually transmission (25.1%; Table 2).

### 3.2. Source of Blood Samples in 714 HIV-Infected Patients that Completed a CD4+ T Cell Test Within One Year of Diagnosis

The sample source of 714 HIV-infected patients completed CD4+ T cell test within one year of diagnosis are shown in Table 3.

### 3.3. Source of Blood Samples in 714 HIV-Infected Patients that Completed a CD4+ T Cell Test Within One Year of Diagnosis

Of 1018 HIV-infected patients, 714 were tested for CD4+ T-cells within one year of diagnosis, accounting for 70.1%. According to the rule that the CD4+ T-cell count was reduced by an average of 50/μL per year, the infection time of ≥8 years accounted for 25.2% (180/714), 5–8 years accounted for 23.2% (166/714), 3–5 years accounted for 18.3% (131/714), and <3 years accounted for 33.2% (237/714).

The average age of infection in the 714 patients was 31.0 ± 13.1 years old. The average time from HIV infection to diagnosis was 4.7 ± 2.4 years. The average infection time of ≥50 years old group was 5.7 ± 2.4 years, which was significantly higher than that of <50 years old group (4.6 ± 2.4 years; *t* = 4.500, *p* < 0.01).

### 3.4. The Recent HIV Infection Rate in HIV-Positive Individuals ≥50 Years Old

A total of 154 HIV-infected individuals ≥50 years old underwent recent infection test, of which, 13.6% (21/154) were diagnosed with a recent HIV infection (infection time is less than 130 days).

### 3.5. The Prevalence of HIV-Positive Individuals Co-Infected with HBV, HCV, and Syphilis

Of 1018 HIV-positive individuals, the rates of HBV, HCV, and syphilis were 11.0%, 11.7%, and 26.0%, respectively. The rates of HBV–HCV, HCV–syphilis, HBV–syphilis, and HBV–HCV–syphilis were 1.7%, 2.2%, 2.6%, and 0.1%, respectively (Figure 1).

### 3.6. Prevalence of HBV Serum Markers

A total of 19 kinds of HBV marker’s phenotype distribution were found. Of which, 43.8% were isolated HBsAb positive (effectively vaccinated against Hepatitis B), and 30.2% were negative for HBV markers (susceptible to Hepatitis B). Moreover, 1.6% were positive for HBsAg, HBeAg, and HBcAb (active Hepatitis B), while 2.9% were positive for HBsAg, HBeAb, and HbcAb (Chronic hepatitis B).

The positive rate of HBsAg in Yi nationality (18.3%) was higher than that of Han nationality (10.5%), and the rate of active acute HBV infection in Yi nationality (5.6%) was significantly higher than that in Han nationality (1.3%; *p* < 0.05).

### 3.7. Risk Factors for HIV–HCV and HIV–Syphilis Co-Infection

The results showed that ethnicity (Yi (OR = 31.030, 95% CI: 11.643–82.694) compared to Han), education level (primary school (OR = 5.393, 95% CI: 1.285–22.633) compared to Illiteracy), and HIV transmission routes (intravenous drug use (OR = 134.024, 95% CI: 14.328–1253.653), intravenous drug use + history of non-married heterosexual contact (OR = 242.534, 95% CI: 10.053–5851.304) and others (OR = 59.696, 95% CI: 6.136–580.787) compared to homosexual transmission) were the risk factors for HCV infection in HIV-positive individuals (Table 4).

The age 31–40 years (OR = 3.423, 95% CI: 1.360–8.615), 41–50 years (OR = 4.809, 95% CI: 1.869–12.375), and >50 years (OR = 2.949, 95% CI: 1.103–7.881) were the higher risk factors and transmission routes compared to <20 years. Compared to homosexual transmission, heterosexual transmission (OR = 0.549, 95% CI: 0.382–0.789) and intravenous drug use (OR = 0.298, 95% CI: 0.112–0.793) were lower risk factors for syphilis infection in HIV-positive individuals (Table 5).

## 4. Discussion

Delayed diagnosis and treatment of HIV, HBV, HCV, and syphilis can lead to severe complications and sequelae including liver cirrhosis, cancer, opportunistic infections, and death [37,38,39]. On the contrary, timely diagnoses and treatment of these co-infections can offset a faster disease progression, reduce transmission to sexual partners and children [40,41,42], and will cumulatively decrease HIV, HBV, HCV, and syphilis epidemics. Hence, it is an important public health issue to assess the rate of these infections in a timely manner among the population. Unfortunately, like HIV-infection, there are no symptoms at the beginning of the infection, and the asymptomatic features of these infections favor their quiet spread. In the absence of knowledge of these diseases, diagnosis, and treatment, they are efficiently spread among vulnerable populations who present risk behaviors [43,44]. According to the results of the first CD4+ T-cell test, the time interval between HIV infection and diagnosis was estimated. It was estimated that 66.8% of HIV-infected patients had been infected with HIV for more than 3 years, and 25.2% had reached more than 8 years. Meanwhile, the source of HIV-infected patients was mainly passive detection such as “pre-operative testing” and “other patient detection”, which was much higher than the active detection method such as voluntary counseling and testing (VCT), indicating that HIV-infected patients were found late [45]. This is consistent with previous experimental results. Because of similar routes of transmission and risk factors, we speculate that HBV, HCV, and syphilis infection among HIV-positive individuals of this study were found late too, as even they were not aware of their infection status at all when they were diagnosed with HIV. The average infection time of ≥50 years old was significantly higher than that of <50 years old group (*t* = 4.500, *p* < 0.01). Only 13.6% of HIV-infected individuals ≥50 years old were diagnosed with recent HIV infection by Limiting antigen avidity enzyme immunoassay (LAg-avidity EIA). The data indicates that late diagnosis is more severe in the elderly population. The phenomenon is likely multifactorial. Subjects ≥50 years old usually do not perceive themselves at risk of HIV infection [46]. Further, shyness and lack of knowledge on how to get a HIV test may cause late detection and late diagnosis [47], and indicate that current sexual health services should be adjusted to better meet their needs.

The epidemiological study on hepatitis B showed that the prevalence of HBsAg in the Chinese population was 7.2% [48,49], and it was 3.5% in the Shaanxi population [50]. In this study, we found that the positive rate of HBsAg for HIV-infected patients in Shaanxi Province was 11.7%, which was higher than the general population in either China or Shaanxi Province. A number of domestic studies have shown that the positive rate of HBsAg in the HIV-infected patients was higher than that of the general population, and it ranged from 13.1% to 19.4% [30,31,32,33]. However, on the contrary, some studies have reported that in HIV-infected patients, the HBsAg positive rate was slightly lower than that of the national general population. He et al. found that the prevalence of HBsAg was 6.3% through a survey which enrolled 1110 cases of HIV-infected patients in central Shanxi, eastern Zhejiang, southwest Yunnan, and northwestern Xinjiang [51]. Ding et al. reported that the positive rate of HBsAg was 5.3% through a retrospective cohort study of HIV-positive individuals receiving combination antiretroviral therapy (cART) between 2004–2016 [34]. The reasons for the inconsistency may be related to many factors such as region, age structure, ethnicity, time of investigation, and transmission routes, indicating that there are certain differences in HBV infection status among HIV-positive individuals in different areas, and further confirmed the necessity of HBV survey in HIV-positive cases of this region.

There are few reports on the five tests for hepatitis B in the HIV/AIDS population in China. A total of 19 kinds of HBV marker’s phenotype distribution were found. Of which, the inoculated population (HBsAg-, HBsAb+, HBeAg-, HBeAb-, HBcAb-) accounted for 43.8%, which was higher than Li’s study on 1314 HIV/AIDS patients (22.5%) [35]. It was close to the hepatitis B vaccination rate of the general population in Shaanxi Province [50], which verified that the hepatitis B vaccination work in Shaanxi Province has achieved certain results. Susceptible populations (HBsAg-, HBsAb-, HBeAg-, HBeAb-, HBcAb-) accounted for 30.2%, similar to Li’s research results (34.4%) [35], which mean that more than 30% of the HIV-infected individuals had not received effective hepatitis B vaccine immunization, and were more susceptible to HBV than the general population. The proportion of HIV-positive individuals with an active acute HBV infection was very low at only 1.6%; however, they could transmit HBV to others very easily. At the same time, the positive rate of HBsAg in Yi nationality (18.3%) was higher than that of Han nationality (10.5%), and the rate of active acute HBV infection in Yi nationality (5.6%) was significantly higher than that in Han nationality (1.3%; *p* < 0.05), suggesting that HBV spread among HIV-positive individuals of Yi nationality in Shaanxi.

Increased liver-related morbidity and mortality among HIV-positive individuals co-infected with HBV [52] indicate the importance of effective hepatitis B vaccine immunization. Ideally, individuals should be inoculated with hepatitis B vaccine before HIV infection as the effectiveness of the vaccination is stronger in HIV-negative individuals [53]. If HIV-positive individuals are not previously vaccinated, they should be vaccinated against HBV when the HIV viral load is low, as HIV viremia can shorten the retention of anti-HBs positive antibodies [54]. Because anti-HBs antibodies can disappear several years after effective HBV vaccination [55], we cannot judge whether individuals negative for anti-HBs in this study were not vaccinated before. Regardless, individuals enrolled in this study were susceptible to HBV, suggesting the strategies to enhance the immune response of immunocompromised individuals are needed. In short, this study has found that the positive rate of HBsAg among HIV-positive individuals was higher than that in the general population. Therefore, the hepatitis B vaccination should be strengthened, especially in HIV-infected patients of Yi nationality. At the same time, the specificity of HIV-infected persons, such as HIV viral load, should be considered.

Unlike HBV infection, there is currently no preventive HCV vaccine and the cost of treatment is enormous. The epidemiological survey showed that the prevalence of anti-HCV was 3.2% in China [56]. In 2016, the epidemiological survey of hepatitis C showed that the anti-HCV positive rate was 0.8% in the general population of Shaanxi Province. In our study, the rate of anti-HCV was 11.1% among HIV-infected patients in Shaanxi Province, which was also higher than in the general Chinese population, and 14-fold higher than in Shaanxi Province. Other studies have found both lower [35] and higher [32] HCV prevalences in HIV-positive patients, ranging from 2.2% to 62.4%. Pelton et al. [57] found that 2357 cases of HIV/AIDS patients in Xinjiang had an anti-HCV positive rate of 38.0%. Chen et al. [32] found that the prevalence of anti-HCV was 62.4% among 978 HIV-infected individuals from Hunan Province, while Li et al. [35] found that the anti-HCV positive rate was only 2.2% in 1314 HIV/AIDS patients under anti-viral treatment in You’an Hospital. The results above were quite different from that in our study. Different results may be related to ethnicity, region, infection route, and investigation time. Multivariate analysis showed that ethnicity, education level, and transmission routes were risk factors resulting in HIV–HCV co-infection. Among 1018 HIV-infected patients, the anti-HCV positive rate of intravenous drug users was as high as 91.5%, which was the risk factor of HIV–HCV co-infection (OR = 134.024, 95% CI: 14.328–1253.653). It has also been reported that the rate of HIV–HCV co-infection in Chinese intravenous drug users is as high as 69.0–93.6%, and the infection rate of other transmission routes is significantly lower than this level [32], indicating that intravenous drug use is the cause of HCV spread in HIV-infected patients in Shaanxi. The results showed a higher prevalence in the Yi nationality than in Han nationality (OR = 31.030, 95% CI: 11.643–82.694). It may be caused by the high rate (41.0%) of intravenous drug use among the Yi nationality in this study. Therefore, the publicity and intervention work should be especially strengthened in intravenous drug users and the Yi nationality. Besides, the ‘others’ group had a higher prevalence (OR = 59.696, 95% CI: 6.136–580.787) compared to homosexual transmission. The ‘others’ group includes blood transfusion or blood product, selling blood for money and mother-to-fetus transmission in this study, which are risk factors for HCV transmission. However, due to strong government control, blood transmission and mother-to-child transmission have been reduced to the lowest level, thus we do not need to worry about “others”.

In recent years, the co-infection of HIV and syphilis has been on the rise in China, especially in MSM. The rate of syphilis was 26.0% in this study, which was the highest of the three monitored diseases, especially as high as 37.1% in MSM. The result was consistent with foreign studies on syphilis epidemics in both HIV-positive MSM and MSM [58,59,60,61]. In China, Li et al. found that the rate of syphilis was as high as 32.8% in HIV-positive MSM [35]. Among the HIV-positive individuals with antibodies of TP, the infection rate of homosexual transmission was so higher but heterosexual transmission was lower (OR = 0.549 95% CI: 0.382–0.789), suggesting that homosexual transmission might be a risk factor resulting in HIV co-infection with syphilis. HIV and syphilis are both sexually transmitted, and lots of people are infected with both pathogens. On one hand, HIV has several effects on the presentation, diagnosis, disease progression, and therapy of syphilis. On the other hand, syphilis can increase the risk of HIV transmission by causing genital ulcers [62]. Therefore, syphilis is a special and important infection because of the morbidity it can enhance the transmission of HIV [63]. In patients co-infected with HIV and syphilis, cutaneous lesions become more severe, symptomatic neuro-syphilis may be more likely to develop, the latency period before the development of meningo-vascular syphilis may be shorter, and the efficacy of standard therapy for early syphilis may be reduced [64]. Moreover, it is worrying that studies show that about 50% of individuals co-infected with syphilis were asymptomatic for syphilis in HIV-positive MSM [65,66]. Therefore, screening for syphilis should be strengthened in both MSM and HIV-positive MSM. In this study, the infection rate of syphilis was 38.8% in college education or above, followed by 26.3% in high school/technical secondary school. Overall, there is no evidence that high education is a risk factor for syphilis infection, through multivariate logistic regression analysis or literature reports. This phenomenon may be caused by the high proportion of MSM in the group with high education.

## 5. Strengths and Limitation

This study is the first large serological survey of HIV-positive patients in Shaanxi to assess co-infections with HBV, HCV, and syphilis. We found that ethnicity and HIV transmission routes were the risk factors for HCV infection in HIV-positive individuals, while homosexual transmission were the risk factors for syphilis infection in HIV-positive individuals. These patients also had the characteristics of low active detection rate and late diagnosis, especially in the elderly. This study provides important information on epidemiology of HBV, HCV, and TP infection in HIV-positive patients in the Shaanxi Province of China.

However, the study also had several weaknesses. First, at baseline tests for CD4+ T-cell counts were not performed on all HIV-infected patients, which restricted our analysis. Second, HIV viral load results at baseline were completely missing. This is an important indicator for evaluating the health status of HIV-infected patients. Third, because the diagnosis of HIV does not require patients to be in a fasting state before testing, fasting has a significant impact on Aspartate aminotransferase (AST) and Alanine aminotransferase (ALT) test results, so as not to see the relationship of liver condition, hepatitis virus infection, and other variables. However, we do not believe these limitations affected our estimates.

## 6. Conclusions

In summary, our study showed that HBV, HCV, and syphilis were wide spread among HIV-positive individuals in Shaanxi Province. The HIV-infected patients in Shaanxi Province had the characteristics of low active detection rate and late diagnosis, especially in the elderly. Moreover, more than 30% of HIV-positive individuals were not effectively vaccinated against HBV, suggesting a need for improved vaccination services.

## Figures and Tables

**Figure 1 ijerph-17-01990-f001:**
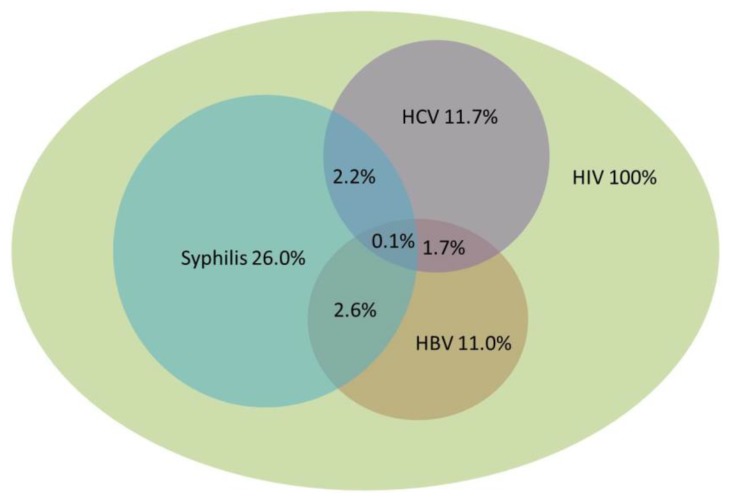
Prevalence of co-infection with Hepatitis B virus (HBV), Hepatitis C virus (HCV), and syphilis in 1018 human immunodeficiency virus (HIV)-infected patients.

**Table 1 ijerph-17-01990-t001:** CD4+ T-cell count level and the estimated human immunodeficiency virus (HIV) infection time and age of population.

CD4+ T-cell Count Level (pieces/μL)	Infection Time (Years)	Infection Age (Years Old)
≤200	≥8	Diagnostic age −8
201–249	5–8	Diagnostic age −7
250–299		Diagnostic age −6
300–349		Diagnostic age −5
350–399	3–5	Diagnostic age −4
400–449		Diagnostic age −3
≥450	<3	Diagnostic age −2

“Diagnostic age” means the age at which AIDS (acquired immune deficiency syndrome) was diagnosed in patients.

**Table 2 ijerph-17-01990-t002:** Characteristics of HBV, HCV, TP in 1018 HIV-infected patients in Shaanxi Province.

Demographic Characteristics	Total HIV (%)	HBsAg (%)	χ^2^	*p*	HCV (%)	χ^2^	*p*	TP (%)	χ^2^	*p*
Overall	1018	112 (11.0)			119 (11.7)			265 (26.0)		
Age (year)			6.710	0.152		5.669	0.225		17.151	<0.01
≤20	56 (5.5)	1 (1.8)			4 (7.1)			6 (10.7)		
21–30	340 (33.4)	37 (10.9)			47 (13.8)			75 (22.1)		
31–40	298 (29.3)	38 (12.8)			37 (12.4)			83 (27.9)		
41–50	183 (18.0)	23 (12.6)			21 (11.5)			65 (35.5)		
>50	141 (13.9)	13 (9.2)			10 (7.1)			36 (25.5)		
Gender			0.087	0.767		0.743	0.389		13.343	<0.01
Male	835 (82.0)	93 (11.1)			101 (12.1)			237 (28.4)		
Female	183 (18.0)	19 (10.4)			18 (9.8)			28 (15.3)		
Ethnicity			4.457	0.108		4.130	<0.01		5.689	0.058
Han	931 (91.5)	98 (10.5)			54 (5.8)			251 (27.0)		
Yi	71 (7.0)	13 (18.3)			61 (85.9)			10 (14.1)		
Others	16 (1.4)	1 (6.3)			4 (25.0)			4 (25.0)		
Marital status			2.956	0.398		15.639	<0.01		1.180	0.758
Unmarried	327 (32.1)	34 (10.4)			32 (9.8)			84 (25.7)		
Married with spouse	476 (46.8)	50 (10.5)			59 (12.4)			120 (25.2)		
Divorce or loss of spouse	205 (20.1)	28 (13.7)			23 (11.2)			59 (28.8)		
Unknown	10 (1.0)	0 (0)			5 (50.0)			2 (20.0)		
Education level			5.654	0.226		1.210	<0.01		21.351	<0.01
Illiteracy	50 (4.9)	9 (18.0)			22 (44.0)			7 (14.0)		
Primary School	161 (15.8)	18 (11.2)			44 (27.3)			45 (28.0)		
Junior high school	442 (43.4)	46 (10.4)			44 (10.0)			97 (21.9)		
High school/technical secondary school	205 (20.1)	27 (13.2)			8 (3.9)			54 (26.3)		
College or above	160 (15.7)	12 (7.5)			1 (0.6)			62 (38.8)		
Transmission routes			6.904	0.233		220.709	<0.01		25.940	<0.01
Homosexual transmission	256 (25.1)	32 (12.5)			5 (2.0)			95 (37.1)		
Heterosexual transmission	667 (65.5)	68 (10.2)			61 (9.1)			155 (23.2)		
Intravenous drug use	47 (4.6)	9 (19.1)			43 (91.5)			9 (19.1)		
Intravenous drug use + history of non-married heterosexual contact	8 (7.9)	1 (12.5)			7 (87.5)			2 (25.0)		
Unknown	21 (2.1)	0 (0)			2 (9.5)			3 (14.3)		
Others #	19 (1.9)	2 (10.5)			1 (5.3)			1 (5.3)		

# Others include blood transfusion or blood product, selling blood for money and mother-to-fetus transmission. HIV: human immunodeficiency virus; HBsAg: Hepatitis B surface antigen represents HBV (Hepatitis B virus) infection; HCV: Hepatitis C virus; TP: Treponema pallidum.

**Table 3 ijerph-17-01990-t003:** Sample source in 714 HIV-infected patients who completed CD4+ T cell test within one year of diagnosis.

Source of Blood Samples	No. of Patients (*n*, %)
STD clinic	20 (2.8)
Positive spouse/sexual partner detection	46 (6.4)
Voluntary Counseling and Testing	148 (20.7)
Other patients/pre-blood/pre-operative tests	268 (37.5)
Thematic survey	56 (7.8)
Blood donation	83 (11.6)
Detainee	35 (4.9)
Others	58 (8.1)

HIV: human immunodeficiency virus; STD: sexually transmitted disease.

**Table 4 ijerph-17-01990-t004:** Univariate and multivariate logistic regression analysis of factors affecting Hepatitis C virus (HCV) infection in 1018 human immunodeficiency virus (HIV)-infected patients in Shaanxi Province.

Variables	β	SE	WALS	*p*	OR (95% CI)
Ethnicity					
Han					1.000
Yi	3.435	0.500	47.174	<0.01	31.030 (11.643–82.694)
Others	1.379	0.824	2800	0.094	3.970 (0.790–19.956)
Marital status					
Unmarried					1.000
Married with spouse	0.364	0.430	0.717	0.397	1.439 (0.620–3.339)
Divorce or loss of spouse	0.697	0.504	1.913	0.167	2.007 (0.748–5.384)
Unknown	2.930	1.049	7806	0.005	18.728 (2.398–146.282)
Education level					
Illiteracy					1.000
Primary School	1.685	0.732	5.301	0.021	5.393 (1.285–22.633)
Junior high school	1.229	0.735	2.796	0.095	3.419 (0.809–14.441)
High school/technical secondary school	0.780	0.854	0.835	0.361	2.182 (0.409–11.639)
College or above		1.351	0.381	0.537	0.434 (0.031–6.130)
Transmission routes					
Homosexual transmission					1.000
Heterosexual transmission	1.948	1.031	3.573	0.059	7.015 (0.931–52.867)
Intravenous drug use	4.898	1.141	18.436	<0.01	134.024 (14.328–1253.653)
Intravenous drug use + history of non-married heterosexual contact	5.491	1.624	11.431	<0.01	242.534 (10.053–5851.304)
Unknown	0.908	1.563	0.338	0.561	2.480 (0.116–53.090)
Others #		1.161	12.410	<0.01	59.696 (6.136–580.787)

β: Beta (standardized regression coefficients); SE: standard error; WALS: weighted-average least squares; OR: odds ratio; CI: confidence interval. *p*: *p*-value. # Others include blood transfusion or blood product, selling blood for money, and mother-to-fetus transmission.

**Table 5 ijerph-17-01990-t005:** Univariate and multivariate logistic regression analysis of factors affecting TP infection in 1018 HIV-infected patients in Shaanxi Province.

Variables	β	SE	WALS	*p*	OR (95% CI)
Age (year)					
≤20					1.000
21–	0.669	0.470	2.029	0.154	1.953 (0.777–4.906)
31–	1.231	0.471	6.830	0.009	3.423 (1.360–8.615)
41–	1.570	0.482	10.605	<0.01	4.809(1.869–12.375)
51–	1.081	0.502	4.648	0.031	2.949 (1.103–7.881)
Gender					
Male					1.000
Female	−0.521	0.246	4.475	0.034	0.594 (0.367–0.962)
Education level					
Illiteracy					1.000
Primary School	0.295	0.481	0.377	0.539	1.343(0.524–3.445)
Junior high school	−0.171	0.468	0.134	0.714	0.843 (0.337–2.108)
High school/Technical secondary school	−0.042	0.491	0.007	0.931	0.959 (0.366–2.508)
College or above	0.504	0.495	1.036	0.309	1.656 (0.627–4.371)
Transmission routes					
Homosexual transmission					1.000
Heterosexual transmission	−0.599	0.185	10.545	<0.01	0.549 (0.382–0.789)
Intravenous drug use	−1.211	0.500	5.872	0.015	0.298 (0.112–0.793)
Intravenous drug use + History of non-married heterosexual contact	−0.408	0.850	0.230	0.631	0.665 (0.126–3.518)
Unknown	−1.302	0.651	4.007	0.045	0.272 (0.076–0.973)
Others #	−1.115	0.671	2.763	0.096	0.328 (0.088–1.221)

β: Beta (standardized regression coefficients); SE: standard error; WALS: weighted-average least squares; OR: odds ratio; CI: confidence interval; TP: Treponema pallidum. *p*: *p*-value. # Others include blood transfusion or blood product, selling blood for money and mother-to-fetus transmission.

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
