# Peer review of "Epidemiological Features and Risk Factors for Acquiring Hepatitis B, Hepatitis C, and Syphilis in HIV-Infected Patients in Shaanxi Province, Northwest China"

_ijerph, 2020, doi:10.3390/ijerph17061990_

Round 1

Reviewer 1 Report

Dear Author,

I have read the paper intitled "Epidemiological Features and Risk Factors for Acquiring Hepatitis B, Hepatitis C and Syphilis in HIV-infected Patients in Shaanxi Province, Northwest China". This paper aims to to investigate the epidemiological features and risk factors of HBV, HCV, and syphilis infection among HIV-positive individuals in Shaanxi Province, which is less developed for economics in northwest China, and their CD4+ T-cell levels at baseline.

In general, the manuscript is well presented, but english grammar should be reviewed. Statistical analysis regarding transmission should be reconsidered. Authors cite homosexual and heterosexual transmission, however it should not be confirmed in this study. This characteristic should be evaluated separately. In addition, authors referred "others", but this was not explained. I recommend to separate these different characteristic since sexual and drug use habits were included in the same setting. Syphilis coinfection was not well discussed, as well, as the impact of education and sexual transmission in this prevalence. 

Reviewer 2 Report

HIV-infected patients are at higher risk for co-infection with Hepatitis B virus (HBV), Hepatitis C virus (HCV), and syphilis, than the general population. In this study Zhang et al analyzed epidemiological data and risk factors for co-infection cases in HIV-positive individuals in Shaanxi Province in China using a retrospective study records. They found that ethnicity and HIV transmission routes were the risk factors for HCV infection in HIV positive individuals while homosexual transmission were the risk factors for syphilis infection in HIV-positive individuals. These patients also had the characteristics of low active detection rate and late diagnosis, especially in the elderly. Moreover, more than 30% of patients were not effectively vaccinated against HBV, suggests a need for improved vaccination services.This study provides an important information on epidemiology of HBV, HCV and TP infection in HIV patients in Shaanxi Province in China.

Overall, I felt the study was well designed and analyzed. When revising, I would like you to focus on the following issues:

Major comments:

It is not clear to me what is the percentage of HIV patients were on ART treatment. This is important because ART treatment may decrease co-infection rates in patients which may confound the interpretation of data. It is better to modify the definition of HCV infection in method section because anti-HCV antibody positive was defined as current or resolved infection. This may lead to different interpretation of data. It is better to include the definition of different HBV markers in 3.6 of result section. Could it be possible for author to include AST and ALT level in the blood to see the relationship of liver condition, hepatitis virus infection and other variables. It is interesting to see if any association between syphilis infection, HIV viral load and CD4+ T cells count in HIV-positive individuals. Ethical approval information is missing.

Minor comments:

Author should discuss the strength and limitation of this study.

Round 2

Reviewer 1 Report

Authors answered the questions raisened and the content of paper was improved.